# Spatiotemporal Distribution and Evolution of Digestive Tract Cancer Cases in Lujiang County, China since 2012

**DOI:** 10.3390/ijerph19127451

**Published:** 2022-06-17

**Authors:** Kang Ma, Yuesheng Lin, Xiaopeng Zhang, Fengman Fang, Yong Zhang, Jiajia Li, Youru Yao, Lei Ge, Huarong Tan, Fei Wang

**Affiliations:** 1Key Laboratory of Earth Surface Processes and Response in the Yangtze-Huaihe River Basin, School of Geography and Tourism, Anhui Normal University, Wuhu 241002, China; 1921011181@mail.ahnu.edu.cn (K.M.); lys1213@mail.ahnu.edu.cn (Y.L.); yaoyouru@ahnu.edu.cn (Y.Y.); gelei@mail.ahnu.edu.cn (L.G.); tanhr@mail.ahnu.edu.cn (H.T.); flynn.wong@mail.ahnu.edu.cn (F.W.); 2Hefei Center for Disease Control and Prevention, Hefei 230022, China; dickyhf@gmail.com (X.Z.); hfcdcljj@163.com (J.L.); 3Department of Geological Sciences, University of Alabama, Tuscaloosa, AL 35487, USA; yzhang264@ua.edu

**Keywords:** digestive tract cancer, incidence, spatiotemporal distribution, clustering, evolution

## Abstract

This study aims to analyze the spatiotemporal distribution and evolution of digestive tract cancer (DTC) in Lujiang County, China by using the geographic information system technology. Results of this study are expected to provide a scientific basis for effective prevention and control of DTC. The data on DTC cases in Lujiang County, China, were downloaded from the Data Center of the Center for Disease Control and Prevention in Hefei, Anhui Province, China, while the demographic data were sourced from the demographic department in China. Systematic statistical analyses, including the spatial empirical Bayes smoothing, spatial autocorrelation, hotspot statistics, and Kulldorff’s retrospective space-time scan, were used to identify the spatial and spatiotemporal clusters of DTC. GM(1,1) and standard deviation ellipses were then applied to predict the future evolution of the spatial pattern of the DTC cases in Lujiang County. The results showed that DTC in Lujiang County had obvious spatiotemporal clustering. The spatial distribution of DTC cases increases gradually from east to west in the county in a stepwise pattern. The peak of DTC cases occurred in 2012–2013, and the high-case spatial clusters were located mainly in the northwest of Lujiang County. At the 99% confidence interval, two spatiotemporal clusters were identified. From 2012 to 2017, the cases of DTC in Lujiang County gradually shifted to the high-incidence area in the northwest, and the spatial distribution range experienced a process of “dispersion-clustering”. The cases of DTC in Lujiang County will continue to move to the northwest from 2018 to 2025, and the predicted spatial clustering tends to be more obvious.

## 1. Introduction

As one of the major diseases seriously threatening human health, cancer has become an important public health and economic problem globally [1,2,3]. As the most populous country in the world, China accounts for more than 23% of new cancer cases globally, and about 50% of new cases are from liver cancer, gastric cancer, and esophageal cancer [3,4]. Digestive tract cancer (DTC) has received widespread attention due to its poor treatment effect and high mortality rate, as well as severe physical, psychological, mental, and economic trauma to patients and their families [5,6,7]. Therefore, research on the traceability and evolution of cancer cases has become particularly important.

In recent years, spatial epidemiological research on the geographic distribution characteristics of cancer has become one of the hotspots of cancer epidemiological research and received extensive attention from the academic community [8,9]. In China, the high-incidence areas of DTC such as liver cancer, gastric cancer, and esophageal cancer have obvious geographic clusters, and the high-incidence areas are mainly distributed in rural areas such as Gansu, Henan, Hebei, Shanxi, and Anhui provinces [10]. However, the previous research was mainly focused on big cities with better medical conditions (such as the cities of Beijing, Shanghai, and Guangzhou), while few research was carried out on rural areas with poor living conditions. Therefore, it is necessary to strengthen the research on the spatiotemporal distribution and evolution characteristics of DTC in rural areas in China.

The current spatial epidemiological research on DTC in China is mainly based on provinces and cities as the research units, and there are relatively few studies on the county scale based on the scale of townships. The current research usually estimates the risk by estimating the crude rate discretely in space, but the i0ncidence of cancer is a small probability event which can be easily affected by factors such as population and region, thus concealing the true situation of the disease [11,12]. Cancer is also a long-term accumulation process with a complex etiology and long incubation period. In the literature, the influence of time was often ignored, leading to certain biases in research conclusions [13]. In addition, the main purpose of cancer spatial epidemiology research is to provide greater clarity for formulating scientific and reasonable specific policies and plans in the follow-up area. Therefore, the spatiotemporal evolution of cancer that has been launched from this should be more worthy of attention.

This study selects Lujiang County, located in the eastern part of Anhui Province, China, as the study site, since it is an area with a high incidence of DTC. From 2010 to 2012, 9064 new cases of malignant tumors were reported in Lujiang County, leading to an average annual crude incidence rate of 245.10 per 100,000. DTC (including liver cancer, gastric cancer, and esophageal cancer) accounted for ~70% of these new cases [14]. DTC has become the largest obstacle to the increase of life expectancy for residents of Lujiang County and has caused a serious burden on economics and society.

The rest of this work analyzes the temporal and spatial distribution, aggregation, and evolution of DTC in Lujiang County, Anhui Province, and then predicts its following evolution at the township scale (Appendix A). A detailed quantification of the spatiotemporal distribution and evolution pattern of DTC in Lujiang County may provide a scientific basis for the formulation of refined and differentiated cancer prevention and control measures in the follow-up area.

## 2. Materials and Methods

### 2.1. Study Area

Lujiang County is located in the central part of Anhui Province, China, between 30°57′~31°33′ N and 117°01′~117°34′ E (Figure 1). It has a humid monsoon climate in the northern subtropics. There are low mountains, hills, polder areas, and lakes in the territory, and the terrain is high in the southwest and low in the northeast. The rivers in the territory belong to the Yangtze River system. The total area of the region is 2343.7 km^2^, with 17 townships and 1 economic development zone under its jurisdiction, and its registered population is approximately 1.2 million. The area is rich in mineral resources such as lead, zinc, copper, and aluminum.

### 2.2. Data Sources and Processing

In this study, residents with household registration in Lujiang County were used as the research object. Based on the registration data of DTC (including liver cancer, gastric cancer, and esophageal cancer here) of residents in Lujiang County from 2012 to 2017, a database of the incidence of DTC was established. The demographic data were sourced from the demographic department, and the case data were sourced from the Center for Disease Control and Prevention in Hefei, Anhui Province, China (http://www.hfcdc.ah.cn/ (accessed on 30 October 2021)). Case data are statistically coded using the International Classification of Diseases (ICD-10). After reviewing the completeness and validity of the data with reference to relevant standards [15], the registration data that meet the standards are selected for statistical analysis. The administrative division map used comes from the resource and environment center data cloud platform (http://www.resdc.cn/ (accessed on 15 November 2021)), and adopts the administrative division of Lujiang County in 2018, including 17 townships and 1 economic development zone. Since the case registration data did not distinguish between the economic development zone and Lucheng town, the administrative divisions of Lujiang County were merged in ArcGis 10.6. The final administrative divisions adopted include 17 townships. The 1:300,000 Lujiang County sub-township layer is selected as the base map data. To ensure accurate overlay analysis of the layers, all layers use the same geographic coordinate system and projected coordinate system, which is GCS_WGS_1984 and WGS_1984_UTM, respectively (with the units of degrees and meters, respectively). The geodetic datum is D_WGS_1984.

### 2.3. Statistical Methods

#### 2.3.1. Spatial Empirical Bayes Smoothing (SEBS) Analysis

Based on the spatial empirical Bayesian smoothing model, K neighborhoods (representing the spatial weights matrix) were defined for each township in Lujiang County. Then the population was accumulated according to the distance from the study area, and finally smoothed according to the rate of the neighboring townships [11,12].

#### 2.3.2. Spatial Autocorrelation Analysis

The spatial correlation strength of DTC in Lujiang County was evaluated by spatial autocorrelation analysis using GeoDa 2.0. The Global Moran’s I index calculated by GeoDa 2.0 software (Chicago, IL, USA) was used to describe the global autocorrelation among all 17 township administrative regions. After 9999 Monte Carlo simulation tests, the standardized statistic Z was used for the statistical test, and the test level was α = 0.05. In general, the Moran’s I index value ranged from −1.0 to +1.0. A Moran’s I index value close to +1.0 indicated that the distribution of cancer patients in Lujiang County is more clustered; whereas a Moran’s I index value close to −1.0 indicated that the distribution of cancer patients in Lujiang County is more discrete. A Moran’s I index value close to zero indicated that the overall distribution of cancer patients is randomly distributed over space without any spatial clustering [15,16,17]. The Local Indications of Spatial Association (LISA) was used to describe the local autocorrelation among all 17 township administrative regions. GeoDa 2.0 software was used to calculate the local spatial autocorrelation index LISA to detect the specific location and type of cancer incidence areas at the township level. Then the identified local space types were exported to ArcGis 10.6 to make a local LISA aggregation map to determine the spatial aggregation type.

#### 2.3.3. Hot Spot Analysis

This tool can identify statistically significant spatial clusters of high values (i.e., hot spots) and low values (i.e., cold spots) using the Getis-Ord G_i_* statistic. Here we applied the G_i_* statistic to estimate the degree of spatial clustering of DTC in Lujiang County. Particularly, the Z value of the hot spot analysis G_i_* statistic was used to identify the cold and hot spots in the distribution of cancer cases among residents in Lujiang County. For example, the value of Z(G_i_*) > 1.96 indicates a high-value spatial cluster or hot spots, while Z(G_i_*) < −1.96 indicates a low-value spatial cluster or cold spot [18,19].

#### 2.3.4. Retrospective Spatiotemporal Scan Statistical Analysis

Kulldorff’s space-time scan statistical analysis was used to explore the spatial and temporal clustering of DTC in Lujiang County from 2012 to 2017 on the township scale. By building a moving cylinder, the radius of the circular window at the bottom varies from 0 to 50% of the total population, and the height corresponds to the study time of the area. The difference in incidence between the inside and outside of the window was calculated [20]. The window with the maximum likelihood was defined as the most likely cluster area, and other windows with statistically significant likelihood ratios (LLR) were defined as the secondary potential clusters. In addition, the relative risk (RR) of the area was calculated, and 9999 Monte Carlo simulations with the test level being a = 0.05 were used to test whether the difference is statistically significant. Since the incidence of cancer is a small probability event, the discrete Poisson probability model was used for scanning. In this study, the maximum spatial scanning area was set to 50% of the total population of Lujiang County, the scanning period was 1 year, and there was no area overlap. We entered the time of onset and the actual number of cases as basic information and then calculated the LLR value and RR value to determine the high-incidence time and high-incidence area. Finally, we used ArcGIS 10.6 to visualize the relative risk of DTC in high-risk cluster areas.

#### 2.3.5. Standard Deviational Ellipse (SDE) Analysis

SDE is a versatile GIS tool for delineating the geographic distribution of the research target. SDE mainly uses the center, major axis, and minor axis as basic parameters to quantitatively describe geographic elements. The center of the SDE reflects the spatial distribution characteristics and relative positions of the ecological elements, and the major and minor axes can reflect the spatial distribution of elements [21,22]. Therefore, the annual metastatic trajectory of the onset of DTC in Lujiang County can be generally revealed.

#### 2.3.6. Grey System Theory Model (1,1) Analysis

The Grey system theory model (1,1), which is a time series forecasting model, has the advantages of simple principles and high prediction accuracy, and can preprocess the original data to obtain better smoothness, making the prediction more effective [23]. This study used the grey GM (1,1) model to predict the incidence of DTC in each township in Lujiang County and explore the future evolution of the spatial pattern of the DTC cases in the county.

## 3. Results and Discussion

### 3.1. Spatiotemporal Distribution of Digestive Tract Cancer

From 2012 to 2017, a total of 14,603 cases of DTC (including 10,605 males and 3998 females) were reported for the residents of Lujiang County. The DTC incidence adjusted by the Chinese standard population was 152.22 per 100,000 (208.79 per 100,000 males, and 88.90 per 100,000 females), and the case number of males is much higher than that of females (Table 1). The incidence of DTC in the county residents is statistically analyzed for each year (Figure 2a). The results show that the average annual incidence rate exhibited an overall downward trend after 2013. The peak incidence occurred in 2012–2013, and the fluctuation decreased after 2013. There is an obvious geographical distribution characteristic of DTC in Lujiang County on a space scale (Figure 2b): the DTC incidence rate generally increased from east to west in the county. The highest incidence was located in the northwest area, especially in Tongda, Tangchi, and Ketan Town.

Compared with the average incidence rate of DTC in Anhui Province (which was 80.45 per 100,000), Lujiang County is an area with a high incidence of cancer, especially gastric cancer (118.19 per 100,000). During the investigation, it was found that the average daily salt intake of the residents in Lujiang County was higher than that in the other areas of Anhui Province, and the residents in the county preferred to eat pickled foods [24]. Studies had also shown that the per capita daily salt intake of the residents in Lujiang County was 8.99 g, and the proportion of residents who consumed pickled food at least 1 time per day accounted for 57.61% of the total population through a follow-up survey of 1392 local residents in Lujiang County [14,25]. High-salt diet and Hp infection are risk factors for gastric cancer, and high-salt diet may enhance the occurrence and development of gastric cancer by regulating Hp gene expression. A synergistic relationship was found between high salt and Hp infection [24,26]. For example, high-salt diet can increase the cluster level of Hp in the stomach, enhance the expression of the toxic factor CagA, and change the viscosity of gastric mucus [10,25,27]. Therefore, the habit of high-salt diet has led to a high overall incidence of DTC in Lujiang County. It is noteworthy that the incidence of DTC in Lujiang County has fluctuated and declined after 2013. This may be affected by the improvement of drinking water and medical and sanitary conditions in the county. Studies showed that after 5–8 years of improved drinking water, the incidence of DTC can be significantly reduced [26,28]. Lujiang County launched a drinking water improvement project in 2009. At the same time, public health (i.e., medical and sanitary conditions) and education improvement also increased the residents’ awareness of health care and might reduce the cancer incidence.

### 3.2. Spatiotemporal Clustering of Digestive Tract Cancer

We applied the spatial autocorrelation analysis method to further analyze the spatial clustering of DTC incidence in different towns. The results showed that Moran’s I value of the DTC incidence in Lujiang County from 2012 to 2017 was 0.491, with the Z scores larger than 1.96 and *p* less than 0.01 (Figure 3), implying that the incidence of DTC in Lujiang County had a significant positive spatial correlation. The clustering characteristic was obvious, and there was an overall spatial dependence. The local autocorrelation analysis was used further to reveal the local distribution characteristics of the DTC incidence. The results showed that there were four clustering modes: high-high clustering, low-low clustering, low-high clustering, and no significant clustering (Figure 4a). In addition, there were obvious high-incidence clustering areas formed spatially, which are mainly distributed in the northwest of Lujiang County, namely Guohe, Jinniu, Tangchi, and Wanshan Town. The hot spot analysis was adopted to further verify the accuracy of the incidence spatial clustering (Figure 4b). Compared to the results of the spatial autocorrelation analysis, the high-incidence clustering areas of cancer detected by the hot spot analysis do not include Wanshan Town. Further analysis found that the morbidity level in surrounding towns was generally higher than that of Wanshan Town. Since high and low values in the local Moran are more likely to attract attention, other clusters around it are ignored, resulting in Wanshan Town being identified as a high-incidence clustering area. Overall, the hot spots of DTC were concentrated in the northwestern area of Lujiang County, which is consistent with the conclusion obtained by the spatial autocorrelation analysis discussed above.

Cancer incidence is not only reflected on the spatial scale, but also in the time scale. To better reveal the temporal evolution of cancer incidence clusters in the geographical space, this study conducted a spatiotemporal scanning analysis of DTC numbers in the entire population of Lujiang County residents from 2012 to 2017 based on the township scale. Kulldorff’s space-time scan statistical analysis using the discrete Poisson model, one most likely high-risk spatiotemporal clustering area and one secondary-risk clustering area were detected in Lujiang County from 2012 to 2017 (Figure 5). Based on the analysis of the RR and LLR of the clustering area, the most likely spatiotemporal cluster area needs more attention. The most likely spatiotemporal clustering area (LLR = 40.82, RR = 1.29, and *p* < 0.001) mainly covered four towns located in the northwestern region of Lujiang County, where the center is Jinniu Town (31.399702° N and 117.203274° E) and the circular area has a radius of 8.23 km (Appendix A). Furthermore, during the high-risk period (2015–2017), a total of 1472 digestive cancer cases were reported in this area. The secondary-risk clustering area is in the southwestern region of Lujiang County (including Leqiao, Luohe, and Nihe Town), and the peak incidence occurred during 2013~2014.

The areas with high incidence of DTC (such as liver, gastric, and esophageal cancer) exhibited obvious regional differences, indicating that they were greatly affected by environmental factors. Many studies confirmed that there is a significant relationship between the ecological environment, population behavior, and the incidence of cancer [29,30,31]. During the field investigation, it was found that the amount of pesticides and fertilizers used in the northwestern region in Lujiang County was significantly higher than the other regions (Appendix A). The extensive use of pesticides and fertilizers caused the accumulation of pollutants in soil and water. These agricultural activities are harmful to human health and increase the risk of DTC [32,33,34]. In addition, industrial activities are closely related to the occurrence of cancer. Studies showed that cancer clusters are significantly related to the number of sewage industrial enterprises [9]. Large industrial enterprises (with an annual output value of more than $20 million) were mainly concentrated in the northwest of Lujiang County, among which included large industrial enterprises such as papermaking, printing, chemicals, and other industries (Appendix A). Pollutants discharged by enterprises entered the human body through the food chain and other channels, causing DTC. The above-mentioned reasons caused the incidence of DTC in the northwest region to be higher than the other regions.

### 3.3. Spatiotemporal Evolution of Digestive Tract Cancer

The standard deviational ellipse model and the grey GM (1, 1) model were adopted to further explore the evolutionary law and future development trend of the DTC incidence in Lujiang County (Appendix A). During the study period, the average center of the incidence of DTC in Lujiang County shifted along the longitude direction from 117.283084° E in 2012 to 117.277296° E in 2017, and in latitude from 31.281476° N in 2012 to 31.290066° N in 2017 (Appendix A). Therefore, the center of gravity of the spatial distribution pattern of DTC in residents of Lujiang County showed a southeast-northwest movement trend from 2012 to 2017, and gradually shifted to the high-incidence area (Figure 6). The total displacement was 0.09 km, of which the westward movement was 0.053 km and the northward movement was 0.037 km (Figure 7a). From 2012 to 2014, it moved generally to the southeast. The area of the standard deviational ellipse during this period increased from 1089.88 km^2^ to 1162.91 km^2^, while the major axis and minor axis increased (Figure 7b), indicating that the incidence of DTC in Lujiang County was spatially expanding from 2012 to 2014, and the distribution tended to be scattered with an obvious spatial spillover effect. From 2015 to 2017, the standard ellipse area was reduced from 1100.97 km^2^ to 1084.32 km^2^, and the major and minor axes were shortened, indicating that the DTC incidence was spatially convergent in Lujiang County from 2015 to 2017. The situation tended to be concentrated in distribution.

Our prediction showed that the center of gravity of the spatial distribution pattern of DTC cases in residents in Lujiang County from 2018 to 2025 will continue to shift to the northwest high-incidence area (Figure 8). The coverage area of the standard deviational ellipse at this stage decreased from 1069.99 km^2^ to 971.34 km^2^, and the major and minor axes will continue to shorten, indicating that for a long period of time in the future, the spatial distribution pattern of DTC in Lujiang County may be converging and shrinking (Appendix A). The agglomeration effect is becoming more obvious (than that before 2018). The above-mentioned analysis shows that with the improvement of basic facilities and the environment in Lujiang County, such as the popularization of tap water, rationalization of the diet structure, continuous improvement of medical and health standards and residents’ health awareness, the incidence rate of DTC is showing a downward trend. However, for a period in the future, the northwestern area of Lujiang County will still be a high-incidence area of DTC, and hence the relevant departments need to emphasize this area as a key area for the prevention and control of DTC.

## 4. Conclusions

Based on geographic information system technology, this study analyzed the spatiotemporal distribution and evolution of digestive tract cancer in Lujiang County, China, from 2012 to 2017. The results suggested that the peak incidence of DTC in Lujiang County occurred in 2012–2013, and the overall spatial distribution pattern increased from east to west. On the spatiotemporal scale, the high-risk spatiotemporal clustering areas were concentrated in the middle and lower reaches of the rivers where the western water system was relatively developed. From 2012 to 2017, the incidence of DTC in Lujiang County showed a southeast-northwest spatiotemporal evolution pattern, and the spatial distribution range experienced a “dispersion-aggregation” change process. The prediction results showed that as the number of DTC patients decreases, the clustering area will move further to the northwest, and the spatial clustering will become more obvious from 2018 to 2025. These findings may shed lights on effective prevention and control of DTC in especially the rural areas in China.

## Figures and Tables

**Figure 1 ijerph-19-07451-f001:**
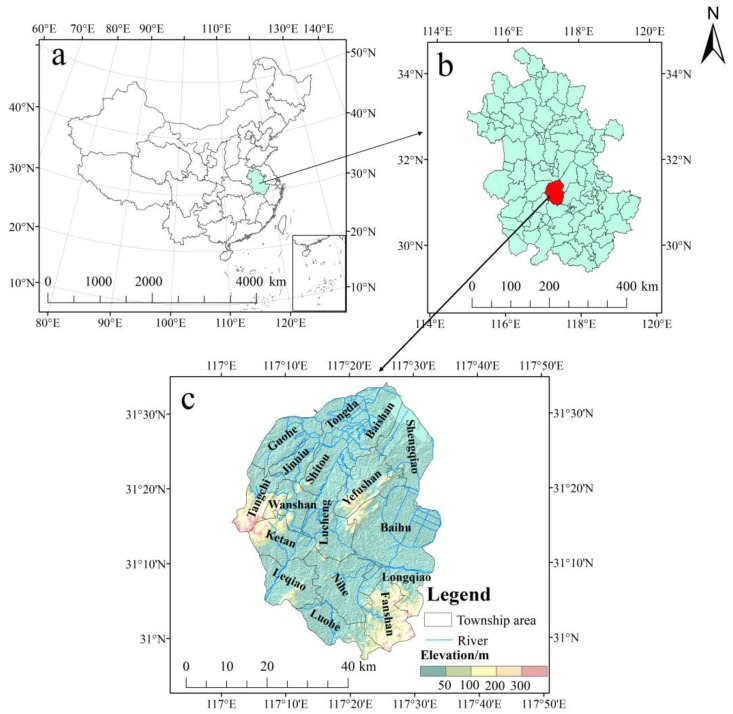
Location of the study area: Lujiang County (**c**) in Anhui province (**b**), China (**a**).

**Figure 2 ijerph-19-07451-f002:**
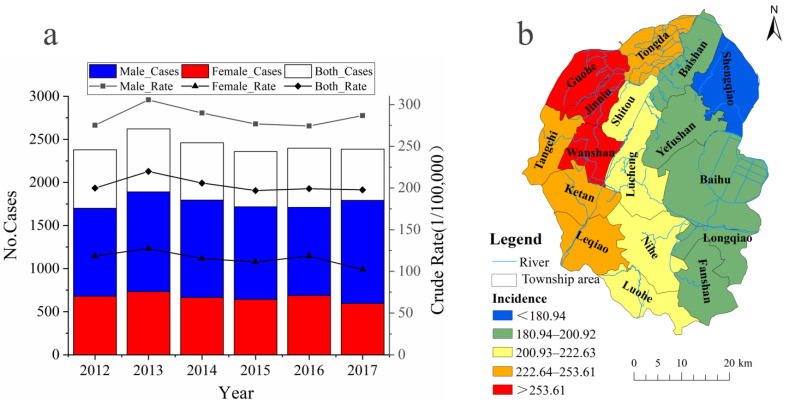
The space (**b**) and time (**a**) distribution of digestive tract cancer in Lujiang County from 2012 to 2017.

**Figure 3 ijerph-19-07451-f003:**
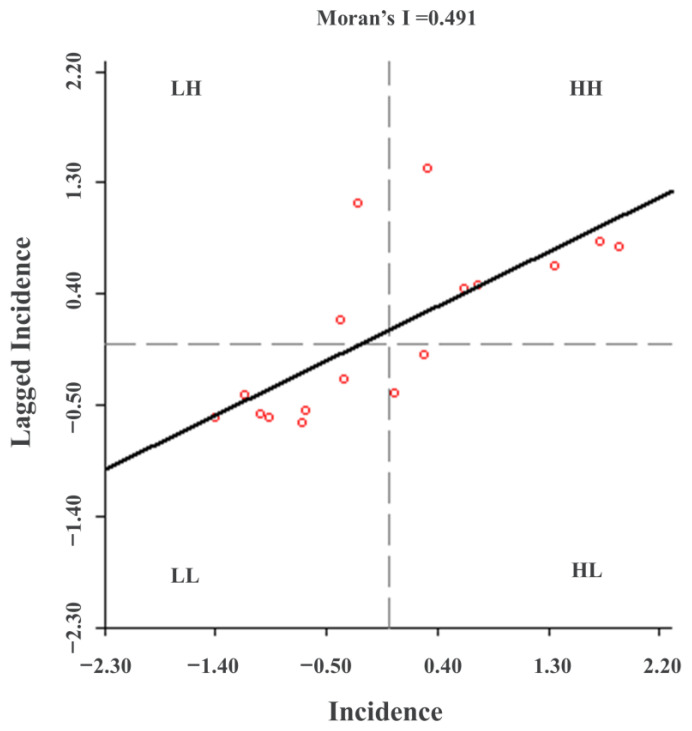
The global Moran’s I spatial autocorrelation analysis of the incidence of digestive tract cancer in Lujiang County from 2012 to 2017. Six high-high agglomeration areas (HH) and seven low-low agglomeration areas (LL) were found using global Moran’s I. (Note: HH stands for high-high agglomeration area, LH stands for low-high agglomeration area, LL stands for low-low agglomeration area, and HL stands for high-low agglomeration area).

**Figure 4 ijerph-19-07451-f004:**
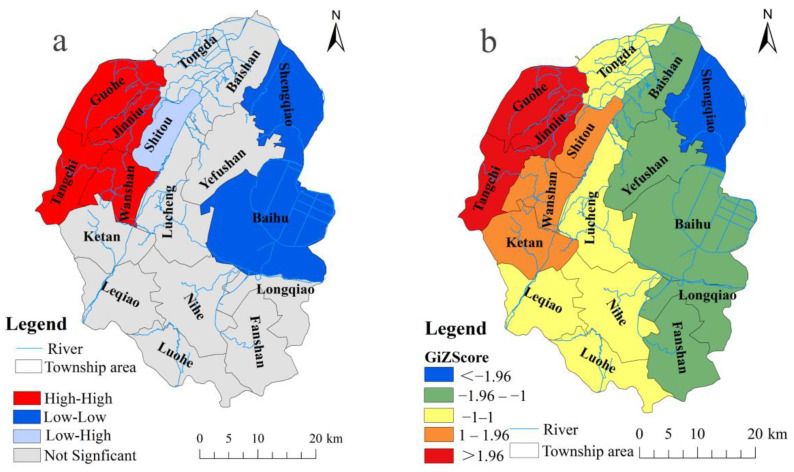
The Local Indications of Spatial Association clusters (**a**) and Local G_i_* hot spot clusters (**b**) of the incidence of digestive tract cancer in Lujiang County from 2012 to 2017.

**Figure 5 ijerph-19-07451-f005:**
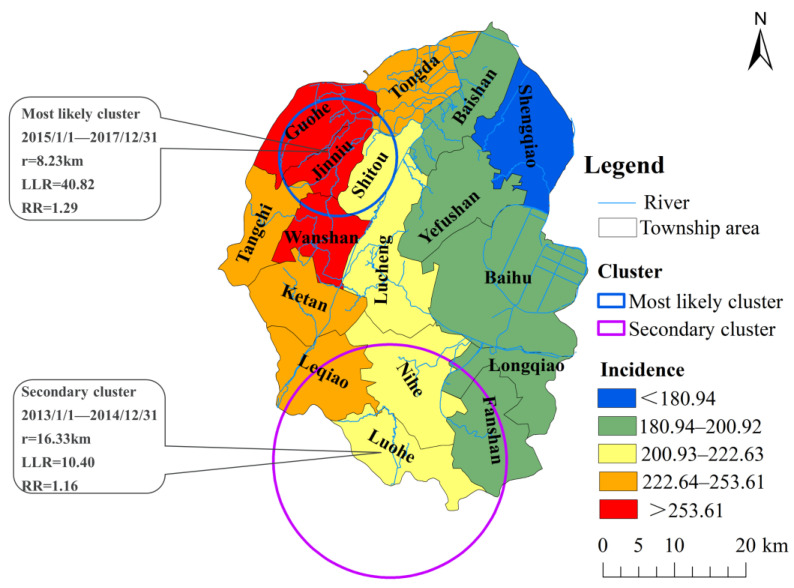
Spatiotemporal clustering of the incidence of digestive tract cancer in Lujiang County from 2012 to 2017. One most likely cluster and one secondary clusters were found using spatiotemporal scanning, which indicated an obvious trend of spatiotemporal clustering for DTC in Lujiang County. The high-risk clusters were predominantly located in the northwest and southwest of Lujiang County.

**Figure 6 ijerph-19-07451-f006:**
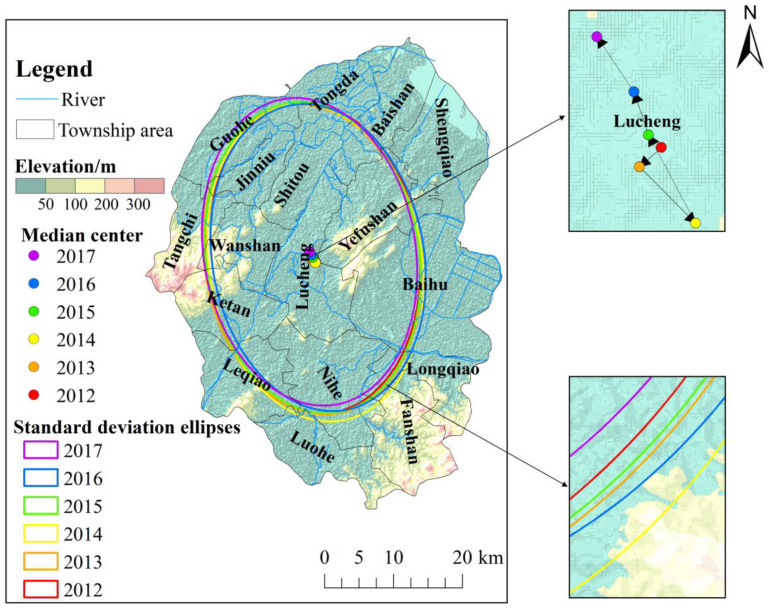
The standard deviational ellipse and gravity center transfer path of the incidence of digestive tract cancer in residents of Lujiang County from 2012 to 2017.

**Figure 7 ijerph-19-07451-f007:**
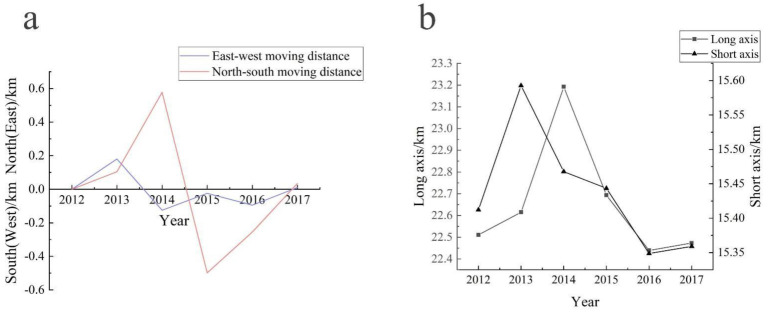
The moving distance (**a**) of gravity center of the digestive tract cancer in residents of Lujiang County and the variation of short and long axis (**b**) from 2012 to 2017.

**Figure 8 ijerph-19-07451-f008:**
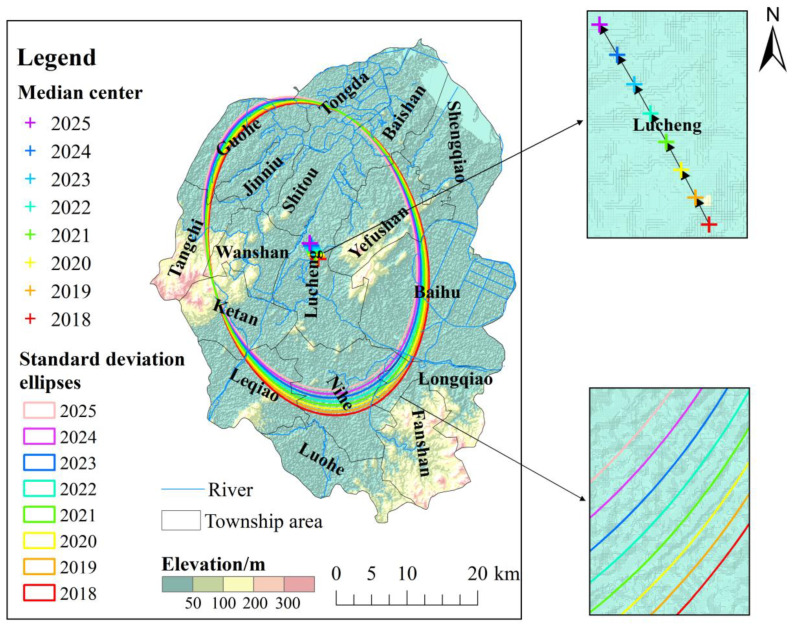
The standard deviational ellipse and gravity center transfer path of the incidence of digestive tract cancer in residents of Lujiang County from 2018 to 2025.

**Table 1 ijerph-19-07451-t001:** Basic situation of digestive tract cancer in residents in Lujiang County from 2012 to 2017.

Year	Male				Female						Both			
	No. Cases	Crude Rate (1/10^5^)	ASR China (1/10^5^)	ASR World (1/10^5^)	No. Cases	Crude Rate (1/10^5^)	ASR China (1/10^5^)	ASR World (1/10^5^)	X^2^ Value ^a^	*p* Value ^a^	No. Cases	Crude Rate (1/10^5^)	ASR China (1/10^5^)	ASR World (1/10^5^)
2012	1700	275.30	202.97	136.10	679	118.47	91.25	58.36	370.19	<0.001	2379	199.81	150.08	119.83
2013	1890	305.69	225.24	151.31	732	127.35	97.39	62.91	434.35	<0.001	2622	219.77	164.64	130.71
2014	1796	289.93	212.24	143.79	664	115.39	88.01	57.07	445.44	<0.001	2460	205.87	153.70	122.16
2015	1717	276.74	202.29	137.46	642	111.17	86.03	55.18	418.20	<0.001	2359	196.92	147.46	116.18
2016	1711	274.41	200.86	136.98	687	118.28	91.08	59.05	369.09	<0.001	2398	199.11	149.10	117.62
2017	1791	286.89	209.14	143.39	594	102.14	79.64	51.05	520.82	<0.001	2385	197.79	148.36	116.59
TOTAL	10605	284.81	208.79	155.92	3998	115.44	88.90	60.30			14,603	203.19	152.22	109.34
APC (%)		−0.50	−0.70	−0.20		−2.86	−2.57	−2.57				−1.00	−1.10	−1.39
*t* Value		−0.43	−0.63	−0.22		−2.10	−2.05	−1.85				−1.16	−1.20	−1.53
*p* Value		0.69	0.56	0.84		0.10	0.11	0.14				0.31	0.30	0.20

Note: ^a^: Chi-square value with Pearson test.

## Data Availability

All data included in this study are available upon request by contact with the corresponding author.

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
