# Peer review of "Spatiotemporal Distribution and Evolution of Digestive Tract Cancer Cases in Lujiang County, China since 2012"

_ijerph, 2022, doi:10.3390/ijerph19127451_

Round 1
Reviewer 1 Report
The document presents important information, however I have some comments:
The title is completely related to the objective, nothing is mentioned regarding the prediction.
It is not clear which is the geographical unit with which the analysis was carried out.
Figure 1 is confusing.
Figures 3, 5 and 6 should be improved, check the correct construction of a map.
The conclusion is very long and confusing, conclude according to the objective.
Reviewer 2 Report
Comments to Manuscript ijerph-1700394_reviewer
The manuscript “ Spatiotemporal distribution and evolution of digestive tract cancer cases in Hefei, China since 2012” are an interesting topic, and the topic of the manuscript is relevant to the journal targeted and interpretation of the results are considered adequate.
1. Abbreviations need to be given their full names when they first appear in a manuscript. GM line 174.
2. What is the main reason causing the DTC in this region?
3. Why is the proportion of men incidence of DTC higher than the women in this area?
4.The conclusion part the author mentioned that the peak incidence of DTC in Lujiang County occurred in 2012-2013, why?
